# Molecular Classification of Endometrial Carcinomas: Review and Recent Updates

**DOI:** 10.3390/cancers18010051

**Published:** 2025-12-24

**Authors:** Anita Kumari, Himani Kumar, Samuel E. Harvey, Deyin Xing, Zaibo Li

**Affiliations:** 1Department of Pathology, The Ohio State University, Columbus, OH 43210, USA; anitak.lhmc@gmail.com (A.K.);; 2Department of Pathology, Johns Hopkins University School of Medicine, Baltimore, MD 21205, USA; sharve18@jhmi.edu (S.E.H.);

**Keywords:** endometrial carcinoma, POLE-ultramutated, microsatellite instability, p53

## Abstract

Recent advances in elucidating the pathogenesis of endometrial carcinoma have highlighted the central role of cancer genomics in refining diagnostic classification, prognostic assessment, and therapeutic stratification. The integration of molecular classification into the latest World Health Organization (WHO) framework underscores the clinical relevance of genomic biomarkers in predicting patient outcomes and informing targeted therapeutic approaches. This review provides a comprehensive overview of the genomic landscape of endometrial carcinoma, emphasizing its significance in advancing precision and personalized medicine.

## 1. Introduction

Endometrial carcinoma (EC) is the sixth most common cancer among women and the second most common malignancy of the female genital tract [1]. Recent cancer statistics indicate that the incidence of EC has increased by more than 1% over the past two decades [2]. In 2025, approximately 69,120 new cases and 13,860 related deaths are projected in the United States [2]. Notably, EC remains one of the few malignancies with continuously rising incidence and mortality rates [2]. The overall 5-year survival rate for EC in the United States is approximately 80% [1].

In 1983, Bokhman proposed a clinicopathologic classification of endometrial carcinoma that categorized tumors into two major groups: Type I and Type II [3]. Type I tumors are estrogen-dependent and predominantly exhibit endometrioid or mucinous morphology. They are commonly associated with metabolic disorders and prolonged estrogen exposure, with endometrial hyperplasia recognized as a precursor lesion. Representing approximately 70% of cases, Type I tumors are typically well to moderately differentiated and carry a relatively favorable prognosis. In contrast, Type II tumors arise in an atrophic endometrium, occur more frequently in postmenopausal and non-obese women, and include serous, clear cell, and other poorly differentiated histologic types. These tumors are characterized by high-grade morphology, aggressive clinical behavior, and an overall poorer prognosis [3].

Bokhman’s landmark study fundamentally revolutionized the classification of EC and standardized its reporting throughout the late 20th century. For decades, this binary system guided diagnostic practice, therapeutic decision-making, and clinical trial design. Subsequent clinicopathologic and molecular investigations have largely validated the diagnostic and therapeutic relevance of this dualistic model; however, several limitations have since become apparent. Notably, poorly differentiated and undifferentiated carcinomas were not adequately encompassed within the original classification, and many high-grade tumors exhibit overlapping morphologic and clinical features that defy strict categorization as Type I or Type II. Moreover, the etiopathogenesis of EC is now recognized to be far more heterogeneous than initially proposed, with substantial biological diversity even within morphologically similar tumors [4]. These observations highlighted the inherent constraints of morphology-based classification systems and underscored the need for a more refined, biologically grounded framework.

Advances in molecular research have demonstrated that tumors with similar histomorphologic features may harbor distinct genetic alterations and exhibit divergent clinical behaviors. The advent of next-generation sequencing (NGS) and large-scale genomic profiling has fundamentally transformed the understanding of endometrial carcinogenesis by The Cancer Genome Atlas (TCGA) [5]. By integrating genomic, transcriptomic, and proteomic data, TCGA provided transformative insights that redefined tumor classification and influenced clinical management. Based on this integrative analysis, TCGA stratified ECs into four distinct molecular subtypes with reproducible genomic signatures and prognostic implications: (a) POLE-ultramutated tumors, characterized by exceptionally high mutational burdens and excellent clinical outcomes; (b) Microsatellite instability (MSI)-hypermutated tumors, associated with defects in mismatch repair and intermediate prognosis; (c) Copy number low (CNL) tumors, largely overlapping with low-grade endometrioid carcinomas and exhibiting relatively favorable outcomes; and (d) Copy number high (CNH) tumors, which include most serous carcinomas and a subset of high-grade endometrioid carcinomas and are associated with extensive chromosomal instability and poor prognosis [5].

Building on these developments, the present review examines the expanding molecular framework of endometrial carcinoma and discusses how these discoveries refine our understanding of disease pathogenesis and inform precision-based clinical management.

## 2. Molecular Classification of Endometrial Carcinomas

The TCGA was a landmark project to redefine endometrial carcinomas by providing molecular insights into its pathogenesis. It shifted EC classification beyond basic histomorphological assessment toward an integrated molecular framework, enabling improved risk stratification and supporting the development of precision oncologic approaches [5]. TCGA was conducted as a comprehensive, multiplatform analysis of 373 endometrial carcinomas including low grade endometrioid, high grade endometrioid and serous carcinomas. Comprehensive molecular analysis including whole exome sequencing, copy number analysis, MSI assays, methylations was carried out on these cases. Based on results, endometrial carcinomas were categorized into four distinct molecular subtypes: (a) POLE-ultramutated, (b) microsatellite instability (MSI)-hypermutated, (c) CNL, and (d) CNH [5].

### 2.1. Polymerase E (POLE)-Ultra-Mutated Subtype

The POLE-ultra-mutated subgroup accounts for approximately 7.3% of endometrial carcinoma (EC) cases [5]. It exhibits the highest mutational burden (~232 × 10^−6^ mutations per megabase) and is therefore termed “ultra-mutated.” The POLE gene encodes the catalytic subunit of DNA polymerase epsilon, a key enzyme involved in DNA replication and repair. Mutations in the exonuclease domain of POLE impair DNA error correction, resulting in an exceptionally high frequency of C → A transversions and producing the characteristic ultra-mutated phenotype of this EC subgroup [5,6,7,8,9].

Clinically, POLE-mutated tumors are associated with excellent outcomes, with progression-free survival rates of 92–100% [5,6,7,8,9]. Compared with other TCGA subtypes, these tumors are more common in younger patients, are associated with lower body mass index, and often present at lower FIGO stages [10]. Over 80% of pathogenic mutations are concentrated in five hotspot residues within the exonuclease domain: Pro286Arg (most frequent), Val411Leu, Ser297Phe, Ala456Pro, and Ser459Phe [9]. Approximately 40% of tumors harbor rare mutations at other exonuclease domain sites, whereas fewer than 4% exhibit mutations outside this domain [9]. See Figure 1 for morphologic and molecular features of POLE-ultra-mutated EC.

Beyond POLE mutations, exome sequencing has identified additional significantly mutated genes (SMGs) with low false discovery rates, including DMD, CSMD1, FAT4, PTEN (94%), PIK3R1 (65%), PIK3CA (71%), FBXW7 (82%), and KRAS (53%) [5,11,12,13]. All POLE-mutated ECs are endometrioid in histology, and nearly 50% present as high-grade tumors, despite their paradoxically favorable prognosis [5]. This discordance suggests that many high-grade endometrioid carcinomas with POLE mutations may be overtreated if management is based solely on morphology [14,15].

Following TCGA, studies have demonstrated that POLE mutations can also occur in undifferentiated/dedifferentiated carcinomas, carcinosarcomas, and clear cell carcinomas [16,17,18,19]. Importantly, POLE mutation status prognostically supersedes p53 and mismatch repair (MMR) status. Even tumors harboring concurrent p53 abnormalities or MMR deficiency retain favorable clinical outcomes when POLE mutations are present [9,20].

Reflecting these findings, the European Society of Gynecological Oncology, European Society for Radiotherapy and Oncology, and European Society of Pathology (ESGO–ESTRO–ESP) guidelines recommend that FIGO stage II POLE-mutated ECs be managed as low-risk tumors [21]. Histologically, these tumors frequently display dense lymphocytic infiltration, likely driven by the high neoantigen load associated with their extreme mutational burden, which may also represent a potential target for immunotherapy [22,23,24]. Recent ESGO–ESTRO–ESP updates integrate the four TCGA molecular subtypes into risk stratification frameworks, refining prognostic assessment and guiding tailored adjuvant therapy recommendations [25].

### 2.2. MSI/Hypermutated Subtype

The TCGA designated this subgroup as the hypermutated subtype due to its elevated mutational load (~18 × 10^−6^ mutations per megabase), second only to the POLE-ultramutated group. This subtype accounts for approximately 28% of EC cases [5]. Mismatch repair deficiency (MMRd) arises from either somatic mutations or epigenetic silencing of key mismatch repair (MMR) genes—MLH1, MSH2, MSH6, and PMS2—which are critical for correcting base-pairing errors during DNA replication. Loss of MMR function results in microsatellite instability (MSI) and a hypermutated tumor phenotype [26]. MSI endometrioid tumors frequently harbor mutations in PTEN, PIK3CA, PIK3R1, ARID1A, RPL22, KRAS, FBXW7, CTNNB1, PPP2R1A, and TP53 [5,11,12,13].

MMR deficiency can be reliably identified by immunohistochemistry (IHC), which serves as a surrogate marker for MMR status. IHC assesses the nuclear expression of MMR proteins, with complete loss of staining in tumor cells—contrasted with retained expression in surrounding stromal or inflammatory cells as an internal positive control—indicating MMR deficiency [27]. The histological patterns and immunohistochemical profile of MMRd tumor subtype are demonstrated in Figure 2.

Similar to POLE-ultra-mutated carcinomas, MMRd ECs are predominantly endometrioid (85.8%), with nearly 47.4% exhibiting high-grade morphology [15]. MMR deficiency is also observed in a subset of undifferentiated and dedifferentiated carcinomas, with a reported frequency of ~44% [16] but is less common in clear cell carcinomas and carcinosarcomas. Importantly, MMRd ECs with serous morphology or immunophenotypic overlap are classified as endometrioid [20,28,29]. Like POLE-mutated tumors, MMRd ECs often display prominent tumor-infiltrating lymphocytes, reflecting their hypermutated and potentially immunogenic phenotype [30].

Clinically, MMRd ECs demonstrate an intermediate prognosis—better than p53-abnormal tumors but less favorable than POLE-mutated cases. Although some MMRd tumors present with high-grade histology, their molecular profile generally corresponds to moderate clinical outcomes [9,31].

According to ESGO–ESTRO–ESP guidelines, MMR-deficient ECs are stratified into risk groups that integrate molecular findings with conventional clinicopathologic parameters, including FIGO grade, myometrial invasion depth, lymphovascular space invasion (LVSI), and histologic subtype [25]. Most MMRd tumors fall within the intermediate-risk category; however, those exhibiting substantial LVSI or deep myometrial invasion may be reclassified as high-intermediate or high-risk [32].

Overall, MMRd ECs exhibit variable prognostic behavior across histotypes. In early-stage, low-grade endometrioid carcinomas, MMR deficiency may paradoxically increase the risk of recurrence [20], whereas high-grade endometrioid tumors with MMR deficiency typically have intermediate outcomes. Interestingly, although uncommon, MMRd status in non-endometrioid carcinomas may confer a better prognosis compared with p53-abnormal counterparts [20,33].

### 2.3. p53 Abnormal/Copy Number High (CNH)/Serous Subtype

Within the TCGA molecular framework, tumors with a low overall mutational load were further divided into two categories: CNH (2.9 × 10^−6^ mutations per Mb) and CNL (2.3 × 10^−6^ mutations per Mb)—based on somatic copy number alteration (SCNA) analysis [5]. Recurrent genetic alterations in these tumors include PIK3CA, PTEN, FBXW7, PPP2R1A, MYC, ERBB2, and CCNE1 [5,11,12,13]. The CNH subgroup largely corresponds to the traditional Type II endometrial carcinomas predominantly comprises tumors with serous morphology (73.3%). These tumors typically arise in older patients, present at advanced stages, and are associated with poor prognosis [20].

P53 abnormal/CNH tumors exhibit a high frequency of TP53 mutations (~85%) and are therefore referred to as p53-abnormal. Immunohistochemistry (IHC) serves as a surrogate marker for p53 status, showing three primary patterns: overexpression, complete loss, and cytoplasmic expression. Overexpression is the most common, observed in 85.6% of tumors and characterized by strong nuclear positivity in 70–80% of tumor cells [20,34]. Complete loss is seen in 11.5% of cases, while cytoplasmic expression is rare, occurring in only 1.9% of tumors [20,34]. (Figure 3) Correlation of p53 status with POLE and MMRd is essential, as both molecular alterations supersede p53 expression in prognostic significance and influence management decisions [20]. Figure 3 shows histomorphological pattern of CNH subtype-serous type of EC. The p53 immunostain (surrogate for TP53 mutations) shows both null and diffuse pattern of staining in mutated tumors.

Histologically, p53 abnormal ECs are higher grade and marked nuclear atypia [29]. While the serous subtype predominates, aberrant p53 expression is also encountered in carcinosarcomas and clear cell carcinomas [18]. These tumors are generally clinically aggressive tumors, demonstrating higher rates of recurrence and poorer overall outcomes compared with other TCGA subtypes [5].

According to the ESGO–ESTRO–ESP guidelines, p53 abnormal ECs are categorized within the high-risk group, with the exception of cases lacking myometrial invasion, which may be managed more conservatively [21].

### 2.4. Copy Number Low (CNL)/Endometrioid/No Specific Molecular Profile (NSMP)

The CNL subtype corresponds to the prototypical Type I EC of the Bokhman classification. These tumors are typically low- to intermediate-grade endometrioid carcinomas, arising in the setting of unopposed estrogen exposure. Among the TCGA molecular subtypes, CNL is the most prevalent, representing approximately 40% of all endometrial carcinoma (EC) cases. These tumors exhibit neither a high mutational burden nor significant copy number alterations, and are generally microsatellite stable (MSS), lacking POLE and TP53 mutations [5]. Recurrent genetic alterations in this group include PTEN, PIK3CA, CTNNB1, ARID1A, and PIK3R1 [5,11,12,13].

The CNL subgroup is heterogeneous, with low-grade endometrioid carcinomas comprising the majority (84.4%) [15]. However, CNL molecular profiles have also been identified in clear cell carcinoma, neuroendocrine carcinoma, high-grade endometrioid carcinoma, undifferentiated/dedifferentiated carcinoma, and carcinosarcoma, though notably not in serous carcinoma [14,16,17,19].

According to ESGO–ESTRO–ESP guidelines, CNL tumors are generally classified as intermediate-risk, although factors such as lymphovascular space invasion (LVSI), tumor grade, and FIGO stage may warrant reclassification to the high-intermediate risk group [25]. Further refinement of prognostic assessment within this subgroup has been suggested by a recent study, which demonstrated that estrogen receptor (ER)-positive ECs exhibit lower recurrence rates and favorable prognosis, even among clinically high-risk patients [35]. Additional biomarkers, including L1 cell adhesion molecule (L1CAM), progesterone receptor (PR), and CTNNB1 (β-catenin) mutations, are under active investigation for their potential role in risk stratification within the CNL/NSMP group [34,36,37].

Momeni-Boroujeni et al. [38] analyzed the molecular profile of NSMP ECs and identified three distinct clusters based on genetic alterations:Cluster 1: Recurrent PIK3R1 and PTEN alterations;Cluster 02: Co-occurrence of PTEN and PIK3CA mutations;Cluster 3: KRAS mutations with 1q gain in the absence of PTEN mutations, or AKT1 mutations with CTNNB1 mutations.

Cluster 3 tumors exhibited higher FIGO grade, advanced stage, and poorer survival compared with clusters 1 and 2. These findings underscore the potential value of further subclassifying NSMP tumors to improve prognostic precision and guide therapeutic strategies [38,39].

## 3. Role of ProMisE and PORTEC Classifier

Two clinically pragmatic molecular classifiers—ProMisE (Proactive Molecular Risk Classifier for Endometrial Cancer) and the PORTEC (Post Operative Radiation Therapy in Endometrial Carcinoma) trial framework—have been instrumental in translating the TCGA molecular classification into routine diagnostic and prognostic practice. These systems utilize surrogate immunohistochemical markers and targeted sequencing to identify molecular subgroups, enabling genomic stratification in standard pathology laboratories [7,10,34,39,40,41].

The ProMisE classifier employs a stepwise algorithm to categorize endometrial carcinomas into the four TCGA-equivalent subgroups. The workflow begins with immunohistochemical evaluation of MMR proteins to identify MMR-deficient tumors, followed by p53 IHC to detect p53-abnormal cases. POLE-ultramutated tumors are then identified through targeted sequencing of the POLE exonuclease domain. Tumors lacking these alterations are classified as NSMP (no specific molecular profile) [7]. This approach provides precise, cost-effective molecular risk stratification, facilitating adjuvant therapy decisions in accordance with ESGO–ESTRO–ESP guidelines, without necessitating comprehensive genomic sequencing [7,42].

Translational PORTEC (TransPORTEC) is a collaborative research consortium focused on ECs that builds on the PORTEC clinical trial program to enhance prognostic stratification and explore novel therapeutic strategies in patients with high-risk ECs. By integrating molecular data with clinicopathologic parameters, TransPORTEC has generated robust evidence linking molecular subtypes to treatment outcomes, thereby refining clinical trial design and supporting the implementation of molecularly guided therapeutic approaches in high-risk EC management [33,34,36].

## 4. Mesonephric-like Adenocarcinoma

Mesonephric-like adenocarcinoma (MLA) was recognized as a distinct tumor type in the 2020 WHO classification of female genital tract tumors [43]. It accounts for roughly 1% of endometrial carcinomas and arises in the uterine corpus, ovary, and para-adnexal soft tissues. In contrast to true mesonephric carcinoma, which derives from Wolffian duct remnants and is typically centered in the cervix, MLA exhibits similar morphologic, immunophenotypic, and molecular profiles yet lacks a demonstrable Wolffian origin [43,44,45]. See Figure 4 for morphologic and immunohistochemical features of MLA. MLA represents an aggressive high-grade endometrial carcinoma with a strong tendency for distant metastasis, particularly to the lungs [43,44,45,46,47].

MLA has only recently been evaluated within the molecular framework of endometrial carcinoma. Molecular profiling shows recurrent KRAS mutations, 1q chromosomal gains, and occasional Müllerian-associated mutations including PIK3CA, PTEN, and CTNNB1. Most tumors fall within the TCGA CNL category, lacking POLE mutations, MMR deficiency, and TP53 abnormalities. A minority of cases, however, show aberrant p53 staining or other molecular deviations and are accordingly assigned to their respective TCGA molecular subgroups based on testing results [43].

## 5. Prognostic Biomarkers in Endometrial Cancers

The prognostic biomarkers are central to the molecular classification of endometrial carcinoma, providing refined risk stratification beyond histopathology and guiding individualized therapy. POLE-ultra-mutated tumors, defined by exonuclease domain mutations, consistently demonstrate excellent outcomes with minimal recurrence, supporting therapy de-escalation [48,49]. MMRd tumors, identified by loss of MLH1, PMS2, MSH2, MSH6, carry intermediate prognosis but predict responsiveness to immune check-point inhibitors [50,51]. P53-abnormal tumors, characterized by TP53 mutations or aberrant p53 immunostaining, are associated with poor outcomes and aggressive clinical behavior, requiring intense therapy [52]. The NSMP group show variable prognosis with CTNNB1 mutation linked to increased recurrence risk while ER/PR positivity favors more favorable prognosis [53,54,55]. Of these markers, only the detection of POLE mutations necessitates genomic analysis. For identifying other molecular alterations—including mismatch repair (MMR) deficiency, abnormal p53 expression, CTNNB1 and ER/PR (IHC) testing is sufficient and can be easily implemented in standard pathology laboratories [41,51]. There are certain emerging biomarkers such as ARID1a mutation, L1CAM expression, Her2 amplification, PDL-1 and tumor infiltrating lymphocytes (TILs) are being investigated for their ability to refine the prognostic assessment and expand the therapeutic options [54,55,56,57,58,59,60]. These markers show that molecular classification is becoming an adaptable tool for precision medicine in EC.

## 6. Application of Molecular Classification in Targeted Therapy

Targeted therapy identifies specific molecular targets within cancer cells that are responsible for tumor growth. This approach represents a recent advancement in EC treatment, driven by substantial research into the molecular pathogenesis of the disease [60]. By focusing on defined oncogenic pathways, targeted therapy aims to inhibit tumor proliferation and progression, offering a more personalized therapeutic approach for patients. As understanding of endometrial carcinoma (EC) biology has advanced, these agents have become an integral component of treatment [60]. Targeted therapy has emerged as a critical component in the management of advanced and recurrent EC, complementing traditional chemotherapy and radiation therapy [60]. In HER2-amplified p53-abnormal tumors and serous carcinomas, the addition of trastuzumab to carboplatin and paclitaxel has demonstrated therapeutic benefit [52,56]. For mismatch repair-deficient (MMRd) tumors, immune checkpoint inhibitors including pembrolizumab and dostarlimab are recommended, while the combination of lenvatinib and pembrolizumab is approved for MMR-proficient recurrent EC [50,51]. Endocrine and mTOR-targeted strategies, such as letrozole with everolimus, are considered in ER-positive NSMP tumors [53,54,55,57]. Rare molecular alterations, including NTRK fusions, can be treated effectively with larotrectinib or entrectinib, highlighting the importance of precision medicine in selected patients [51,58,59].

## 7. Future Prospective

Looking ahead, the integration of comprehensive molecular profiling into routine clinical practice promises to further refine risk stratification and enable truly personalized therapeutic approaches. Advances in next-generation sequencing, liquid biopsy, and multi-omics analyses are expected to uncover new biomarkers and therapeutic targets, as highlighted in recent reviews [51,54,55,60]. The continued evolution of clinical trials designed around molecular subtypes will be critical in realizing the full potential of precision oncology in endometrial carcinoma.

Despite its transformative impact, molecular subtyping of ECs remains associated with important limitations and ongoing controversy. Molecular categories do not always align with traditional histopathologic classification or tumor grade, as some morphologically low-grade tumors fall into aggressive molecular groups and vice versa, prompting debate over whether molecular features should supersede or complement morphology in routine practice. For example, most carcinosarcoma tumors (morphologically high-grade) fall within the p53-mutant and NSMP categories, with rare cases classified as POLE-mutated or MMR-deficient subtypes [61]. However, a study reported that all carcinosarcomas were exclusively p53-mutant, as tumors previously categorized as non–p53-mutant were reclassified as endometrioid variants after second review [62]. Additional challenges include intratumoral heterogeneity, temporal evolution of molecular alterations, and potential sampling bias, particularly when classification is based on limited biopsy material [63,64]. Practical implementation of TCGA-derived and surrogate classifiers is further complicated by interlaboratory variability in IHCs and sequencing methodologies, leading to occasional discordant subtype assignment [34,65,66]. Finally, disparities in cost, access, and standardization for IHC/molecular testing (especially POLE mutation testing) limit uniform adoption across regions and/or institutions, underscoring the need for harmonized testing, prospective validation, and refined clinical guidelines to fully realize the benefits of molecular classification in EC [67].

## 8. Conclusions

The TCGA molecular classification has transformed the understanding and management of endometrial carcinoma by providing an objective system that goes beyond traditional histopathology. By defining four molecular subtypes and integrating key prognostic biomarkers—such as POLE mutations, mismatch repair deficiency, and p53 abnormalities—this framework enables more accurate risk stratification and precision-guided therapy. As molecular profiling becomes increasingly integrated into clinical practice, it is set to further enhance individualized patient care. Ongoing research and molecularly stratified clinical trials will be crucial to fully harness the potential of precision oncology and improve outcomes for patients with endometrial carcinoma.

## Figures and Tables

**Figure 1 cancers-18-00051-f001:**
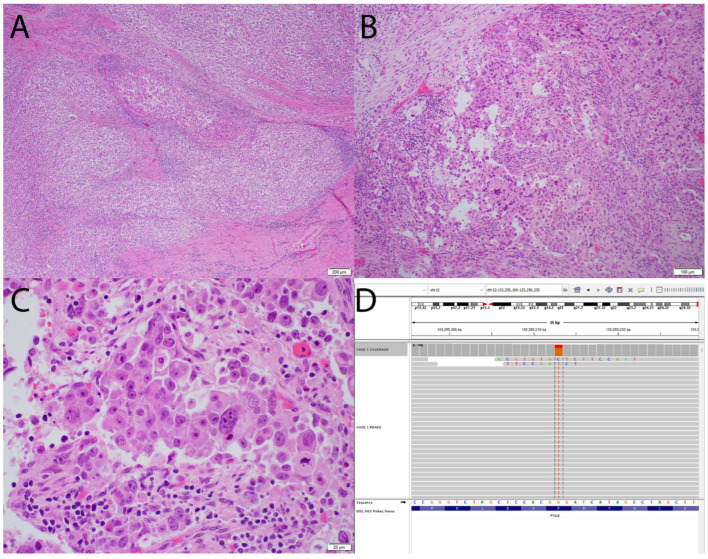
Morphologic and molecular characteristics of a POLE-ultra-mutated endometrial carcinoma. (**A**–**C**) The tumor shows solid architecture, marked pleomorphism, brisk tumor-infiltrating lymphocytes, and atypical mitoses—features typical of POLE-ultramutated tumors ((**A**): 40×, (**B**): 100×, (**C**): 400×, H&E stain). (**D**) Integrated genomics viewer (IGV) demonstrates a heterozygous exonuclease-domain POLE mutation (chr12:133,250,210 C>T).

**Figure 2 cancers-18-00051-f002:**
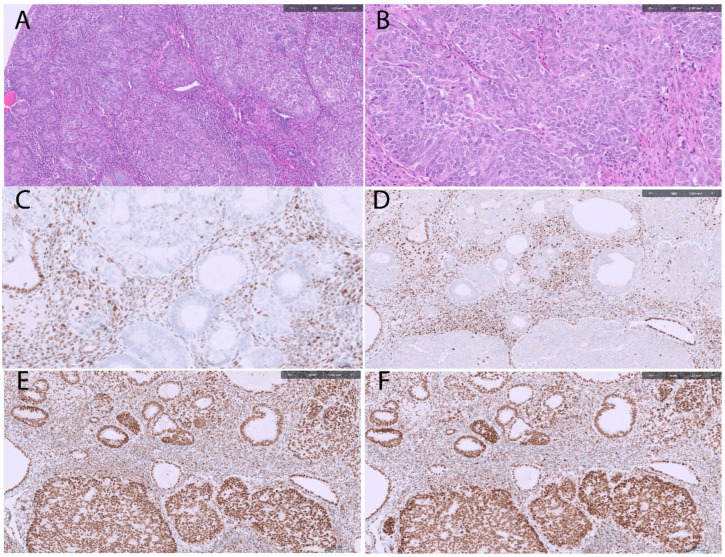
MMR deficient FIGO 2 endometrioid carcinoma. (**A**,**B**) The tumor shows glandular and solid architecture with moderate nuclear atypia (100×, 200×, H&E stain). (**C**,**D**) Tumor cells show loss of MLH1 and PMS2 expression with positive internal control (nuclear staining) ((**C**): MLH1 immunostain, (**D**) PMS2 immunostain, 100×). (**E**,**F**) The tumor shows intact expression of MSH2 and MSH6 (nuclear staining) ((**E**): MSH2 immunostain; (**F**): MSH6 immunostain,100× IHC).

**Figure 3 cancers-18-00051-f003:**
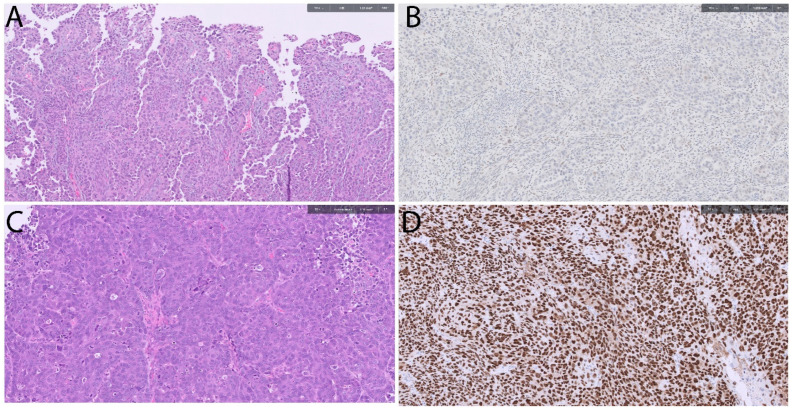
Endometrial serous carcinoma. (**A**) The tumor shows papillary and solid pattern and high-grade nuclear atypia (H&E stain, 100×). (**B**) The tumor shows mutated (null) staining for p53 (no staining) (Immunostain, 100×). (**C**) The tumor shows predominantly solid pattern and high-grade nuclear atypia (H&E stain, 100×). (**D**) The tumor shows mutated (diffuse nuclear staining) for p53 (Immunostain, 100×).

**Figure 4 cancers-18-00051-f004:**
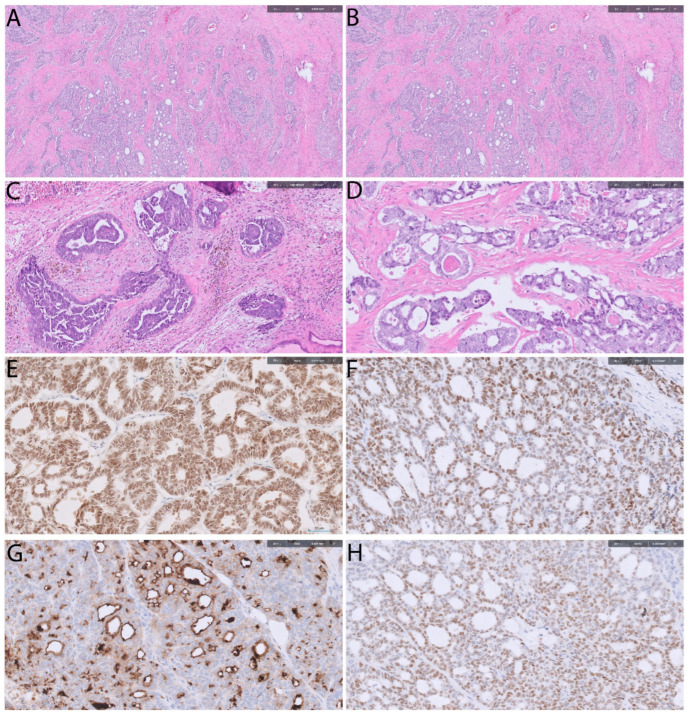
Mesonephric-like adenocarcinoma. (**A**,**B**) Solid, tubular, retiform and glandular patterns (H&E stain, 100× and 200×). (**C**,**D**) Glomeruloid structures and the hallmark tubular architecture with intraluminal eosinophilic secretions (H&E stain, 100× and 200×). Tumor cells are diffusely positive for PAX8 (nuclear staining) (**E**), focally positive for TTF-1 (nuclear staining) (**F**), CD10 (Membranous and cytoplasmic staining) (**G**), and GATA3 (nuclear staining) (**H**) (Immunostain, 100×).

## Data Availability

No new data were created or analyzed in this study. Data sharing is not applicable to this article.

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
