# Peer review of "Molecular Classification of Endometrial Carcinomas: Review and Recent Updates"

_cancers, 2025, doi:10.3390/cancers18010051_

Round 1
Reviewer 1 Report
Comments and Suggestions for Authors
The review article on 'Molecular Classification of Endometrial Carcinomas' is a significant contribution to the field. Authors have explained the different molecular types of endometrial cancer. However, there are certain concerns with this manuscript. Please see the following and revise the manuscript as required.
1) Authors mentioned in the methods section of the abstract that they conducted a systematic literature review. In such case, they have to re-format the whole manuscript as a systematic review. They have to include the period of literature search, what were the search terms used for this literature search, how many studies were found for each search term and how did they choose the final articles to be included in the review. Either the authors have to re-format the manuscript as a systematic review or they have to change it as a regular review article.
2) In the introduction section, the rationale and objective of this review is not clear.
3) The last paragraph of the introduction could be added under a main section, say, Section 2 as 'molecular classification of EC' and they could mention briefly about the TCGA project as well as different molecular categories of EC and each of the molecular subtypes could be re-numbered as sub-sections, 2.1, 2.2, 2.3 and so on.
4) Figures1-4, authors mentioned that these are images captured using tumour tissue samples. The manuscript lacks information regarding the sample details and ethical approval. The authors are requested to provide these in accordance with the journal’s guidelines.
5)When a quick PUBMED search with the term 'Molecular Classification of Endometrial Carcinomas" was conducted, 21 systematic reviews and 271 review articles were found. Information from some of those articles could be added in this review. Please see the following.
- Raffone A et.al., Gynecol Oncol. 2021 Aug;162(2):401-406.
- Marchetti M, et.al., Gynecol Oncol. 2025 Sep;200:145-154.
- Pizzimenti C,et.al.,. Cancers (Basel). 2025 Jul 22;17(15):2420.
- Kim HK, Kim T. Cells. 2025 Sep 8;14(17):1404.
- Secord AA, Powell MA, McAlpine J. Obstet Gynecol. 2025 Nov 1;146(5):660-671.
- Kaya M, et.al., Int J Gynecol Cancer. 2025 Jun;35(6):101759.
6) The manuscript may be enhanced by providing further details on the application of this molecular classification in personalized medicine, illustrated with more examples.
Author Response
The review article on 'Molecular Classification of Endometrial Carcinomas' is a significant contribution to the field. Authors have explained the different molecular types of endometrial cancer. However, there are certain concerns with this manuscript. Please see the following and revise the manuscript as required.
1) Authors mentioned in the methods section of the abstract that they conducted a systematic literature review. In such cases, they have to re-format the whole manuscript as a systematic review. They have to include the period of literature search, what the search terms used for this literature search, how many studies were found for each search term and how did they choose the final articles to be included in the review. Either the authors have to re-format the manuscript as a systematic review, or they have to change it as a regular review article.
Response: The article has been changed to regular review article instead of systemic literature review. The abstract has been modified according to a review article.
2) In the introduction section, the rationale and objective of this review is not clear.
Response: Changes have been made to introduction part such that rationale and objective are clearly stated.
3) The last paragraph of the introduction could be added under a main section, say, Section 2 as 'molecular classification of EC' and they could mention briefly about the TCGA project as well as different molecular categories of EC and each of the molecular subtypes could be re-numbered as sub-sections, 2.1, 2.2, 2.3 and so on.
Response: The above-mentioned portion of the article has been structured and re-numbered according to the suggestions made.
4) Figures1-4, authors mentioned that these are images captured using tumor tissue samples. The manuscript lacks information regarding the sample details and ethical approval. The authors are requested to provide these in accordance with the journal's guidelines.
Response: Relevant text added to the respective figures within the manuscript in addition to the figure legends. There is no need to of ethical approval as all images are from de-IDed cases without ant PHI (Patient’s Health Information).
5) When a quick PUBMED search with the term 'Molecular Classification of Endometrial Carcinomas" was conducted, 21 systematic reviews and 271 review articles were found. Information from some of those articles could be added to this review. Please see the following.
Raffone A et.al., Gynecol Oncol. 2021 Aug;162(2):401-406.
Marchetti M, et.al., Gynecol Oncol. 2025 Sep;200:145-154.
Pizzimenti C,et.al.,. Cancers (Basel). 2025 Jul 22;17(15):2420.
Kim HK, Kim T. Cells. 2025 Sep 8;14(17):1404.
Secord AA, Powell MA, McAlpine J. Obstet Gynecol. 2025 Nov 1;146(5):660-671.
Kaya M, et.al., Int J Gynecol Cancer. 2025 Jun;35(6):101759.
Response: The suggested articles and additional latest articles have been added to manuscript and in the references.
6) The manuscript may be enhanced by providing further details on the application of this molecular classification in personalized medicine, illustrated with more examples.
Response: Separate paragraphs added to the manuscript mention about biomarkers and personalized medicine in EC.
Reviewer 2 Report
Comments and Suggestions for Authors
Reviewer Comments:
- Motivated by the expanding role of genetic testing in cancer care, the authors analyze genetic test data from patients with Endometrial Carcinomas.
- A preliminary analysis would be useful as an example of how this proposed protocol can be applied in a systematic review.
- Add references that provide a concise overview of traditional genomics, including foundational methods such as DNA sequencing, gene expression profiling, and variant analysis. These references should help contextualize how conventional genomic approaches have been used to study disease mechanisms and guide clinical decision-making, and how they relate to or differ from the methods proposed in this work.
- For the data availability section, the authors should specify which molecular and proteomic data are available, in addition to the current statements describing what data are not available.
Author Response
Motivated by the expanding role of genetic testing in cancer care, the authors analyze genetic test data from patients with Endometrial Carcinomas.
A preliminary analysis would be useful as an example of how this proposed protocol can be applied in a systematic review.
Add references that provide a concise overview of traditional genomics, including foundational methods such as DNA sequencing, gene expression profiling, and variant analysis. These references should help contextualize how conventional genomic approaches have been used to study disease mechanisms and guide clinical decision-making, and how they relate to or differ from the methods proposed in this work.
For the data availability section, the authors should specify which molecular and proteomic data are available, in addition to the current statements describing what data are not available.
Response: The article has been changed to regular review article instead of systematic literature review article. A new paragraph is added to the introduction part mentioning about the use of genomic testing methods used in TCGA project.
Reviewer 3 Report
Comments and Suggestions for Authors
- The article successfully bridges molecular biology with clinical practice, emphasizing how molecular classification informs prognostic stratification and personalized therapeutic approaches.
- The review adequately covers all four TCGA molecular subtypes (POLE-ultramutated, MSI-hypermutated, copy-number low, and copy-number high), including their distinct genomic alterations, histopathologic features, and clinical behaviors
- The inclusion of recent ESGO-ESTRO-ESP guidelines demonstrates the authors' commitment to providing current clinical perspectives.
- The discussion of ProMisE and PORTEC classification systems demonstrates practical translation of molecular classification into routine diagnostic pathology
- The abstract could be more concise and better highlight the key findings and clinical implications.
Some sections contain formatting inconsistencies, particularly in citation numbering and reference presentation. - The systematic literature review methodology requires more detailed description, including specific databases searched, inclusion/exclusion criteria, and time frame of literature coverage.
- While the manuscript references guidelines updated to 2025, it would benefit from inclusion of more recent clinical trials (2023-2025) validating molecular classification in therapeutic decision-making.
- Consider adding a summary table comparing the four molecular subtypes with their key genetic alterations, histological features, clinical behaviors, and therapeutic implications.
9. The section on mesonephric-like adenocarcinoma, while informative, appears somewhat disconnected from the main focus on TCGA classification. Better integration into the molecular framework is recommended.
10. Ensure consistent formatting of genetic nomenclature (e.g., POLE vs. POLE-ultramutated).
11.Some figures references in text (e.g., Figure 1, 2, 3, 4) should be properly cited and described in the figure legends.
12.Some cited references appear to be from earlier periods (pre-2020). Updating with more recent publications (2022-2025) would strengthen the evidence base.
Author Response
The article successfully bridges molecular biology with clinical practice, emphasizing how molecular classification informs prognostic stratification and personalized therapeutic approaches.
The review adequately covers all four TCGA molecular subtypes (POLE-ultramutated, MSI-hypermutated, copy-number low, and copy-number high), including their distinct genomic alterations, histopathologic features, and clinical behaviors
The inclusion of recent ESGO-ESTRO-ESP guidelines demonstrates the authors' commitment to providing current clinical perspectives.
The discussion of ProMisE and PORTEC classification systems demonstrates practical translation of molecular classification into routine diagnostic pathology.
1. The abstract could be more concise and better highlight the key findings and clinical implications.
Response: The abstract has been modified to be concise highlighting key features and clinical implications.
2. Some sections contain formatting inconsistencies, particularly in citation numbering and reference presentation.
Response: Sections formatting and structuring have been done accordingly.
3. The systematic literature review methodology requires more detailed description, including specific databases searched, inclusion/exclusion criteria, and time frame of literature coverage.
Response: The article has been changed to regular review article instead of systemic literature review. The abstract has been modified according to a review article.
4. While the manuscript references guidelines updated to 2025, it would benefit from inclusion of more recent clinical trials (2023-2025) validating molecular classification in therapeutic decision-making.
Response: Latest reference articles have been added to the manuscript.
5. Consider adding a summary table comparing the four molecular subtypes with their key genetic alterations, histological features, clinical behaviors, and therapeutic implications.
Response: A summary table has been added to the manuscript comparing the key features of molecular subtypes.
6. The section on mesonephric-like adenocarcinoma, while informative, appears somewhat disconnected from the main focus on TCGA classification. Better integration into the molecular framework is recommended.
Response: We completely agree with the reviewer’s comment. The following paragraph was added into the revised manuscript to address the integration of mesonephric-like adenocarcinoma into the molecular framework.
MLA has only recently been evaluated within the molecular framework of endometrial carcinoma. Molecular profiling shows recurrent KRAS mutations, 1q chromosomal gains, and occasional Müllerian-associated mutations including PIK3CA, PTEN, and CTNNB1. Most tumors fall within the TCGA copy-number low/no specific molecular profile category, lacking POLE mutations, MMR deficiency, and TP53 abnormalities. A minority of cases, however, show aberrant p53 staining or other molecular deviations and are accordingly assigned to their respective TCGA molecular subgroups based on testing results.
7. Ensure consistent formatting of genetic nomenclature (e.g., POLE vs. POLE-ultramutated).
Response: Consistent formatting of POLE-ultramutated has been done.
8. Some figures references in text (e.g., Figure 1, 2, 3, 4) should be properly cited and described in the figure legends.
Response: Relevant text added to the respective figures within the manuscript in addition to the figure legends.
9. Some cited references appear to be from earlier periods (pre-2020). Updating with more recent publications (2022-2025) would strengthen the evidence base.
Response: Latest reference articles have been added to the manuscript.
Round 2
Reviewer 1 Report
Comments and Suggestions for Authors
Authors have revised the manuscript substantially. Majority of the suggestions were considered by them.
For the 4th comment,
4) Figures1-4, authors mentioned that these are images captured using tumor tissue samples. The manuscript lacks information regarding the sample details and ethical approval. The authors are requested to provide these in accordance with the journal's guidelines.
Author's Response: Relevant text added to the respective figures within the manuscript in addition to the figure legends. There is no need to of ethical approval as all images are from de-IDed cases without ant PHI (Patient’s Health Information).
Review: The figures,1-4 show H&E stain as well as include immunohistochemistry images. Please include the methods in detail for this as a supplementary section. Authors could label the staining (cell surface/cytoplasmic/nuclear) by using an arrow in each IHC image.
Author Response
Review: The figures,1-4 show H&E stain as well as include immunohistochemistry images. Please include the methods in detail for this as a supplementary section. Authors could label the staining (cell surface/cytoplasmic/nuclear) by using an arrow in each IHC image.
Response: Added a supplementary section with H&E stain and IHC methods. The staining pattern (membranous/cytoplasmic/nuclear) was indicated in the figure legend.
Reviewer 3 Report
Comments and Suggestions for Authors
1 Significant text duplication is observed between the Abstract and Introduction (e.g., lines 19–25 vs. 37–42). Phrases like "endometrial carcinoma remains one of the leading causes of gynecologic cancer-related mortality" appear verbatim in both sections, reducing conciseness and readability
2 Inconsistent naming of molecular subtypes: "CNL" and "copy number low" are used interchangeably without standardization
3 References 1–3 are listed multiple times without corresponding in-text citations, creating confusion
4 Several key statements (e.g., incidence rates, risk factors) lack supporting references or cite placeholder entries (e.g., "[21]" with no corresponding reference).
5 The review largely summarizes existing knowledge without sufficient discussion of limitations, contradictory evidence, or unresolved debates in molecular subtyping
6 Multiple typographical errors and incomplete sentences (e.g., lines 60–61: "arises from the endometrial lining of the uterusis sixth most common cancer…"
7 Figures 1–4 are referenced in the text, but the captions include placeholder descriptors (e.g., "Figure 1. Morphologic and molecular characteristics…" appears editorial rather than finalized). Ensure figures are explicitly tied to subsections (e.g., "See Figure 1 for POLE-mutated EC histology")
Author Response
Reviewer 3:
1 Significant text duplication is observed between the Abstract and Introduction (e.g., lines 19–25 vs. 37–42). Phrases like "endometrial carcinoma remains one of the leading causes of gynecologic cancer-related mortality" appear verbatim in both sections, reducing conciseness and readability.
Response: The duplicated text between abstract and introduction was changed.
2 Inconsistent naming of molecular subtypes: "CNL" and "copy number low" are used interchangeably without standardization.
Response: Changed names and made them consistent.
3 References 1–3 are listed multiple times without corresponding in-text citations, creating confusion.
Response: References 1-3 were cited in text.
4 Several key statements (e.g., incidence rates, risk factors) lack supporting references or cite placeholder entries (e.g., "[21]" with no corresponding reference).
Response: Reference [21] (Concin N, Matias-Guiu X, Vergote I, Cibula D, Mirza MR, Marnitz S, Ledermann J, Bosse T, Chargari C, Fagotti A, et al. ES-GO/ESTRO/ESP guidelines for the management of patients with endometrial carcinoma. Int J Gynecol Cancer. 2021 Jan;31(1):12-39.) was cited in page 6 and 7.
5 The review largely summarizes existing knowledge without sufficient discussion of limitations, contradictory evidence, or unresolved debates in molecular subtyping.
Response: More discussion about limitations, contradictory evidence and unresolved debates in molecular subtyping was added.
6 Multiple typographical errors and incomplete sentences (e.g., lines 60–61: "arises from the endometrial lining of the uterusis sixth most common cancer…"
Response: Typographical errors were corrected.
7 Figures 1–4 are referenced in the text, but the captions include placeholder descriptors (e.g., "Figure 1. Morphologic and molecular characteristics…" appears editorial rather than finalized). Ensure figures are explicitly tied to subsections (e.g., "See Figure 1 for POLE-mutated EC histology").
Response: Changed.